# Changes in Multiparametric Magnetic Resonance Imaging and Plasma Amyloid-Beta Protein in Subjective Cognitive Decline

**DOI:** 10.3390/brainsci13121624

**Published:** 2023-11-23

**Authors:** Qiaoqiao Xu, Jiajia Yang, Fang Cheng, Zhiwen Ning, Chunhua Xi, Zhongwu Sun

**Affiliations:** 1Department of Neurology, The First Affiliated Hospital of Anhui Medical University, Hefei 230022, China; xqq7039@126.com (Q.X.); yjj1005@126.com (J.Y.); 2Department of Neurology, The Third Affiliated Hospital of Anhui Medical University (Hefei City First People’s Hospital), Hefei 230061, China; 13355679962@163.com (F.C.); nnn0628@126.com (Z.N.)

**Keywords:** subjective cognitive decline, gray matter volume, percent amplitude offluctuation, graph theory, plasma amyloid-beta protein

## Abstract

The association between plasma amyloid-beta protein (Aβ) and subjective cognitive decline (SCD) remains controversial. We aimed to explore the correlation between neuroimaging findings, plasma Aβ, and neuropsychological scales using data from 53 SCD patients and 46 age- and sex-matched healthy controls (HCs). Magnetic resonance imaging (MRI) was used to obtain neuroimaging data for a whole-brain voxel-based morphometry analysis and cortical functional network topological features. The SCD group had slightly lower Montreal Cognitive Assessment (MoCA) scores than the HC group. The Aβ42 levels were significantly higher in the SCD group than in the HC group (*p* < 0.05). The SCD patients demonstrated reduced volumes in the left hippocampus, right rectal gyrus (REC.R), and right precentral gyrus (PreCG.R); an increased percentage fluctuation in the left thalamus (PerAF); and lower average small-world coefficient (aSigma) and average global efficiency (aEg) values. Correlation analyses with Aβ and neuropsychological scales revealed significant positive correlations between the volumes of the HIP.L, REC.R, PreCG.R, and MoCA scores. The HIP.L volume and Aβ42 were negatively correlated, as were the REC.R volume and Aβ42/40. PerAF and aSigma were negatively and positively correlated with the MoCA scores, respectively. The aEg was positively correlated with Aβ42/40. SCD patients may exhibit alterations in plasma biomarkers and multi-parameter MRI that resemble those observed in Alzheimer’s disease, offering a theoretical foundation for early clinical intervention in SCD.

## 1. Introduction

With aging and dietary adjustments, an increasing number of individuals with normal cognition are expressing concerns about memory impairments, known as subjective cognitive decline (SCD), which renders them more likely to progress to dementia than those without such concerns [1,2,3]. SCD is believed to be an early stage in the continuum of Alzheimer’s disease (AD), potentially occurring over 10 years before the onset of mild cognitive impairment (MCI) [4,5,6]. Individuals with SCD perform well on neuropsychological tests but subjectively experience cognitive decline, implying that the evaluators’ subjective judgments greatly influence their classifications. Hence, in response to these inconsistencies, Jessen et al. proposed diagnostic criteria for AD-related SCD [7]. They also defined certain characteristics of AD-related SCD, including the presence of the apolipoprotein E (APOE/ε4) genotype and evidence of AD-related plasma biomarkers. Additionally, they added two supplemental features for SCD, including a long duration of SCD and seeking medical help owing to SCD [8].

There is a lack of consensus regarding the findings of multiparametric magnetic resonance imaging (MRI) in patients with SCD. Patients with SCD exhibit a significant decrease in fractional anisotropy and a significant increase in mean diffusivity, particularly in the hippocampus and entorhinal cortex [9,10]. However, the studies conducted by Kiuchi et al. and Viviano et al. did not indicate statistically significant differences in the diffusion metrics [11,12]. Some studies have shown that patients with SCD exhibit reduced volume in the olfactory cortex and hippocampus compared to healthy individuals, with a yearly volume decline of 1.5% relative to the baseline level [13,14,15]. Other studies have demonstrated that patients with SCD exhibit decreased gray matter volume (GMV) in the hippocampus and frontal–temporal lobes, which is associated with a decline in memory cognitive scores [16,17]; however, other research reports did not find such significant changes [9,18]. Previous studies have employed resting-state functional MRI (rs-fMRI) to analyze the amplitude of low-frequency fluctuations (ALFF) and network topological characteristics in patients with SCD [19,20]. The default mode network (DMN) plays an important role in processing episodic memory, self-referential processing, social cognition, and overall brain function [21]. It is composed of a highly interconnected set of brain regions, including the posterior cingulate cortex, medial prefrontal cortex, lateral temporal cortex, and hippocampus [22]. The current research on SCD predominantly centers on variations within the intrinsic functional network of the DMN, yielding inconsistent findings. Verfaillie et al. [23] found increased connectivity between the posterior DMN and the medial temporal memory system in SCD patients; on the contrary, another study found decreased connectivity between the DMN and the hippocampus [24]. Although there are inconsistencies between their findings, all studies suggest the important role of the DMN in SCD, and further research with more rs-fMRI data is needed.

Some studies have shown that in SCD patients, the ALFF of spontaneous brain activity is higher in the left inferior parietal lobule and lower in the precuneus and cerebellum [25]. Changes in the ALFF are related to the verbal episodic memory scores of SCD patients [19]. When the ALFF and some ALFF features are combined, the accuracy in distinguishing between SCD patients and healthy individuals is higher compared to that when using a single feature [25]. Additionally, persistent atrial fibrillation (PerAF) has been reported to be more reliable and sensitive than ALFF and fractional ALFF during fMRI analysis [25,26]. PerAF represents the percentage of the blood-oxygen-level-dependent (BOLD) signal fluctuation relative to the average BOLD signal intensity at each time point, averaged over the entire time series [27]. However, the use of the PerAF method to assess SCD is currently limited. This study focused on the characteristic changes of PerAF in SCD.

Additionally, the association between plasma amyloid-beta protein (Aβ) and SCD remains controversial. With an improved test accuracy, plasma Aβ measurements, particularly the total plasma 42/40 ratio, are reported to improve the identification of SCD patients with concomitant brain amyloidosis and reduce screening failures in preclinical AD studies. They can also reduce the number of amyloid positron emission tomography (PET) scans by 62%, significantly decreasing the recruitment time and costs [28,29]. Therefore, we conducted multiparametric MRI examinations in this study to explore the correlation between neuroimaging findings, plasma Aβ, and neuropsychological scales. Our hypotheses were as follows: (1) the Aβ levels are higher in individuals in the SCD group than those in the healthy control (HC) group; (2) patients with SCD exhibit AD-like structural changes in cognitively related areas; (3) changes in the PerAF and network topological characteristics are more prominent in individuals in the SCD group than in those in the HC group; and (4) changes in the cortical structure, PerAF, and brain network topological characteristics are associated with plasma biomarkers Aβ and Montreal Cognitive Assessment (MoCA).

## 2. Materials and Methods

### 2.1. Ethical Considerations

This study was approved by the ethics committee of the First Affiliated Hospital of Anhui Medical University and was conducted in accordance with the Declaration of Helsinki. All participants provided written informed consent to publish their personal and clinical data and medical images.

### 2.2. Study Design and Participants

We included 53 patients with SCD and 46 age- and sex-matched HCs. The inclusion criteria for the SCD group were as follows: (1) right-handedness, (2) aged between 55 and 75 years, (3) reported SCD, and (4) neuropsychological assessment results including MoCA scores > 19, >22, and >24 for elementary school education or below, middle school education, and university education, respectively; Mini-Mental State Examination (MMSE) scores > 17, >20, and >24 for illiterate individuals, elementary school education, and junior high school education or higher, respectively; Cambridge Cognitive Examination-Chinese version (CAMCOG-C) scores ≥ 90, and Clinical Dementia Rating (CDR) score of 0. The exclusion criteria were as follows: (1) severe cerebrovascular disease; (2) other diseases or factors that may cause cognitive decline, such as physical illnesses, immune abnormalities, thyroid dysfunction, depression, history of brain injury, psychiatric history, drug dependence, and alcohol intoxication; (3) left-handedness; and (4) inability to cooperate with MRI scanning, plasma collection, and cognitive testing. The inclusion criteria for the HC group were as follows: (1) no evidence of cognitive impairment and (2) no brain atrophy or apparent white matter hyperintensities observed in cranial imaging.

### 2.3. Clinical Data Collection and Neuropsychological Scale Assessment

Detailed information of all participants, including age, sex, years of education, height, weight, and general clinical data, such as hypertension, diabetes, hyperlipidemia, smoking history, and alcohol history, were collected. Two neurologists who received professional training conducted neuropsychological scale assessments on the participants, including the MMSE, MoCA, CDR, Cambridge Cognitive Examination-Chinese version (CAMCOG-C), and Activity of Daily Living scale (ADL), to evaluate all participants’ overall cognitive function and daily living abilities. The MMSE evaluates orientation, memory, calculation, attention, recall, and language. MoCA is a brief (approximately 10 min) screening tool for MCI that evaluates visual space, executive function (clock drawing), naming, attention, language, abstract ability, memory, and orientation with a total score of 0–30. A higher MMSE and MoCA score represents better cognitive function. The ADL scale, which has been revised by Professor Zhang Mingyuan, is a Chinese version of the scale consisting of 20 items that are answered by the subject or an informed person. A total score of 20 indicates complete normality, while a score greater than 20 suggests varying degrees of functional decline, with 80 being the highest achievable score. Individual ability analysis is divided into two levels: a score of 1 indicates normality, while scores of 2–4 indicate functional decline. The CAMCOG-C scale consists of the cognitive section of the Cambridge Examination for Mental Disorders of the Elderly, which includes multiple subtests covering aspects such as attention, memory, executive function, language, and visuospatial abilities. The total score ranges from 0 to 105, with a normal range being ≥ 90 points. It can effectively differentiate between normal cognitive function and MCI [30]. Meanwhile, the CDR scale has emerged as a crucial instrument in longitudinal studies and clinical diagnoses for assessing the stages and severity of AD. Currently, it continues to be frequently employed in clinical trials.

### 2.4. Plasma Aβ42 and Aβ40 Detection

Fasting plasma samples were collected from all participants within 1 week of the MRI scan. The plasma samples were collected in purple tubes containing ethylenediaminetetraacetic acid or sodium citrate as an anticoagulant. The tubes were centrifuged at 3000 rpm for 20 min, and the upper plasma was carefully collected into EP tubes and stored at −80 °C to prevent repeated freezing and thawing. The plasma samples were tested for Aβ42 and Aβ40 concentrations using enzyme-linked immunosorbent assay (ELISA). This assay kit uses a sandwich method with two antibodies to measure the levels of Aβ42 or Aβ40 in the samples. Purified human Aβ42 or Aβ40 antibodies were coated on a microplate to form a solid-phase antibody. Aβ42 or Aβ40 was added to the coated wells, followed by the addition of horseradish peroxidase (HRP)-labeled Aβ42 or Aβ40 antibodies, forming an antibody–antigen–enzyme-labeled antibody complex. After thorough washing, the 3,3′,5,5′-tetramethylbenzidine (TMB) substrate was added for color development. TMB turns blue under the HRP enzyme’s catalysis and turns yellow under acidic conditions. The intensity of the color is positively correlated with the levels of Aβ42 or Aβ40 in the sample. This method utilizes the specific reaction between antigens and antibodies, while enzyme technology relies on the color reaction between enzymes and substrates to display the binding of antigens and antibodies. This greatly enhances the detection sensitivity, making the detection level comparable to radioactive assays. The colored products generated through enzyme reactions are mostly stable, which is beneficial for sample preservation. The absorbance (OD value) is measured at a wavelength of 450 nm using an ELISA reader, and the concentration of human Aβ42 or Aβ40 in the sample is calculated through a standard curve, resulting in more accurate results (Wuhan Huada Biological Technology Co., Ltd., Wuhan, China).

### 2.5. APOE Genotyping

For the APOE gene genotyping, 500 μL of mixed plasma samples from the participants was extracted into EP tubes and stored at −80 °C. Genomic DNA was extracted from the plasma samples using the plasma DNA Mini Kit (DP348) from China BeiJing TianGen Biochemical Technology Co., Ltd., Beijing, China. The rs429358 and rs7412 (APOE) gene genotyping was performed using a five-primer amplification refractory mutation system. After the polymerase chain reaction, the plates were read using a TEcan M1000 reader, and the DNA sequences were analyzed using the online software, SNP decoder1 (China BeiJing LiuHe HuaDa Gene Technology Co., Ltd., Beijing, China).

### 2.6. MRI Scanning and Processing

Image acquisition: MRI scanning was performed using a 3.0T MR scanner (Discovery MR750w, General Electric, Milwaukee, WI, USA) in the Imaging Department of the First Affiliated Hospital of Anhui Medical University. A 24-channel head coil was used to acquire the three-dimensional (3D) high-resolution brain volume T1-weighted imaging and T2 fluid-attenuated inversion recovery (T2 FLAIR) sequences in the sagittal plane. The participants were instructed to remain awake and quiet and to keep their heads as still as possible. The specific parameters used were as follows:(1)Structural MRI scanning: T1-weighted 3D-SPGR sequence imaging. TR = 9.5 ms, TE = 3.9 ms, TI = 450 ms, flip angle = 20°, and matrix size = 512 × 512.(2)rs-fMRI scanning: SE-EPI sequence imaging. TR = 2 ms, TE = 30 ms, FOV = 240 × 240 mm, flip angle = 90°, matrix size = 64 × 64, slice thickness = 4 mm, and slice gap = 0.6 mm.

### 2.7. Preprocessing of fMRI Data and the Network Topology Feature

The data preprocessing was conducted using Statistical Parametric Mapping 12 (SPM12, London, UK http://www.fil.ion.ucl.ac.uk/spm (accessed on 12 August 2023)) on the MATLAB platform. The rs-fMRI data processing assistant software (DPARSF, http://rfmri.org/dparsf (accessed on 12 August 2023)) (DPABI 4.3, http://rfmri.org/dpabi (accessed on 12 August 2023)) was employed for data preprocessing. Each participant’s first 10 volumes of functional data were discarded, followed by (http://rfmri.org/dparsf (accessed on 12 August 2023)) temporal and head motion correction. Subsequently, all images were normalized to the standard Montreal Neurological Institute template and resampled to a resolution of 3 × 3 × 3 mm. To minimize the impacts of head motion and non-neuronal BOLD fluctuations, a set of 24 head motion parameters and average signals from the white matter, cerebrospinal fluid, and global signals were employed as nuisance covariates. Finally, the detrending process was applied to eliminate the offset caused by the sensor during data processing.

Brain networks are typically compared with random networks to test whether they are configured with significantly non-random topology. We constructed random networks using the GRETNA software. The random networks are generated using a Markov wiring algorithm, which preserves the same number of nodes and edges and the same degree of distribution as the real brain networks. The data in this study were converted into a binary network for analysis. For a binary network, two edges, (i1,j1) and (i2,j2), are first selected at random; that is, node i1 is connected to node j1, and node i2 is connected to node j2. If there are no edges between node i1 and node j2 and between node i2 and node j1, we then add two new edges, (i1,j2) and (i2,j1), to replace the original two edges, (i1,j1) and (i2,j2). This procedure is repeated 2 × the number of edges in the reference brain network to ensure randomized organization [31]. Specifically, automated anatomical labeling atlas was used to divide the brain into 90 cortical and subcortical regions of interest, and each region was considered a network node. Next, the mean time series was calculated for each region, and partial correlations of the mean time series between all pairs of nodes (representing their conditional dependences by excluding the effects of the other 88 regions) were considered as the edges of the brain functional connectome. This process resulted in a 90 × 90 partial correlation matrix for each subject, which was converted into a binary matrix (i.e., adjacency matrix) according to a predefined threshold, where the entry aij = 1 if the absolute partial correlation between regions i and j exceeds the threshold. Otherwise, aij = 0. The number of random network computations is 100 times. To ensure a more scientifically rigorous approach and prevent excessively isolated nodes, we first utilized the functions provided by the Gretna 2.0 software to determine the minimum sparsity value for participants, which was found to be 0.047. A sparsity that is too large is considered a non-biological network, so the maximum sparsity is set to 0.3. Subsequently, we calculated the data for 26 network sparsity values ranging from 0.05 to 0.3 (with an interval of 0.01) and examined the degree centrality data. We discovered isolated nodes at network sparsity values of 0.05–0.11. Therefore, in this study, the range of sparsity values is set between 0.12 and 0.3, with an interval of 0.01. Finally, we calculated the global properties for the remaining 19 network sparsity values.

The small-world coefficient (σ) can determine the small-world attribute. σ = γ/λ, where the normalized clustering coefficient (γ) = Creal/Crandom and the normalized shortest path length (λ) = Lreal /Lrandom (real and random represent the actual and random network, respectively). If the network under study has an Lp value close to that of the matched random network and a larger Cp value, it has the characteristics of a small-world network, such as a small-world coefficient, σ > 1, indicating that it possesses the characteristics of a small-world network; otherwise, it does not. In this study, we calculated the average values of small-world properties (aSigma) and the average global efficiency (aEg) at 19 thresholds of network sparsity. Subsequently, these mean values were compared for inter-group analysis.

### 2.8. Preprocessing of Structural MRI Data

The CAT12 toolbox, which is based on SPM12 (see http://www.fil.ion.ucl.ac.uk/spm (accessed on 12 August 2023)), was used to conduct a voxel-based morphometry analysis. The general process was as follows:

Segmentation: T1-weighted structural images were segmented into gray matter, white matter, cerebrospinal fluid, and other tissues using tissue probability maps. Normalization: The segmented brain tissues were spatially normalized. Jacobian modulation: The GMV was modulated using the Jacobian determinant generated during spatial normalization. Smoothing: The GMV density map modulated via Jacobian modulation was further smoothed to reduce spatial noise. The smoothing kernel (full-width half maximum) was selected as 6 × 6 × 6 mm.

First, the image quality of each patient was manually assessed, and those who did not meet the standards were excluded, including obvious signal loss, artifacts, head motion artifacts, and other apparent image issues. The weighted average image quality rating (IQR) was used as the criterion for evaluating image quality, and patients with a weighted average (IQR) of <70% were excluded.

### 2.9. Statistical Analyses

To validate hypothesis 1, the demographic and clinical characteristic differences between patients in the SCD and HC groups were analyzed using a chi-squared test and independent sample *t*-test in IBM SPSS Statistics for Windows, version 22.0 (IBM, Armonk, NY, USA). A further logistic regression analysis was performed to determine the statistical significance of the differences, using a significance threshold of *p* < 0.05.

To validate hypotheses 2 and 3, the independent sample *t*-test was performed to examine the differences in PerAF and GMV between patients in the SCD and HC groups. Population characteristics, including total intracranial volume, sex, age, and education level, were incorporated as covariates in the group analysis. The statistical threshold was set at *p* < 0.05 (family-wise error (FWE)/false discovery rate (FDR) correction).

Graph theory analysis was conducted on the brain functional networks of the two groups to observe changes in global and nodal properties. The statistical threshold was set at *p* < 0.05 (FDR correction).

To validate hypothesis 4, regions showing significant differences in the above analysis were further analyzed for their correlations with serum markers (Aβ42, Aβ40, and Aβ42/40) and neuropsychological scales. The significance level was set at *p* < 0.05 (FDR correction).

## 3. Results

### 3.1. Comparison of General Clinical Data

(1) The demographic characteristics and clinical data of the two groups are presented in Table 1. This study included 53 patients with SCD and 46 matched controls. The MoCA score of the SCD group was slightly lower than that of the HC group, and the Aβ42 levels were significantly higher in the SCD group than in the HC group (*p* < 0.05).

(2) APOE allele and genotype comparisons: Three and two participants in the HC and SCD groups had missing genetic results, respectively. The chi-squared tests did not demonstrate statistically significant differences between the two groups in the APOE allele and genotype distributions (*p* > 0.05) (Table 2 and Table 3).

### 3.2. GMV Results

In patients with SCD, the GMV decreased in the left hippocampus (HIP.L), right rectal gyrus (REC.R), and right precentral gyrus (PreCG.R), with *p* = 0.040, *p* = 0.037, and *p* = 0.039, respectively (FWE correction) (Figure 1). Within the group of patients with SCD, the correlation analysis revealed a significant positive correlation between MoCA and GMV in the HIP.L, REC.R, and PreCG.R. In contrast, the volume of the HIP.L was negatively correlated with Aβ42, and that of the REC.R was negatively correlated with Aβ42/40 (Figure 2).

The red regions indicate areas where patients in the SCD group have decreased GMV compared to those in the control group. The histograms represent the comparison of mean ± standard error of the mean for the differential regions in both groups.

### 3.3. PerAF Comparison Results

Compared to healthy individuals, patients with SCD exhibited increased PerAF in the left THA, with *p* = 0.042 (FDR correction) (Figure 3). A regression analysis within the group of patients with SCD revealed a negative correlation between the MoCA scores and PerAF in the left THA (Figure 4).

The brain regions marked in red represent increased PerAF values in the patients in the SCD group compared to those in the HC group. The histograms represent the comparison of the mean ± standard error of the mean for the PerAF values in both groups.

### 3.4. Graph Theory Analysis

The patients in the SCD group showed significantly lower levels of aSigma and aEg than those in the control group (*p* < 0.01 and *p* = 0.013, respectively) (FDR correction) (Figure 5). aSigma was positively correlated with the MoCA scores, and aEg was positively correlated with Aβ42/40 (Figure 6).

The comparison of the median and interquartile ranges is depicted using violin plots.

## 4. Discussion

Cerebrospinal fluid (CSF) Aβ42/40 (or CSF Aβ42) and Aβ-PET are commonly used as biomarkers to detect the Aβ pathology in the brain and may work as early biomarkers related to cognitive dysfunctions [32]. However, the collection of CSF through lumbar puncture is often viewed as a more invasive diagnostic method, while Aβ-PET scans can be costly and pose potential risks of gamma radiation exposure for both medical professionals and patients. Recently, with the rapid development of highly sensitive detection, it has become possible to measure brain-specific protein levels in the plasma, providing the best opportunity for the successful treatment and prevention of cognitive decline before clinical symptoms appear [33]. Research has demonstrated a strong correlation between the plasma Aβ42/40 ratio and cerebral amyloidosis [28]. In a study investigating SCD associated with AD, plasma Aβ42/40 has been confirmed as a reliable biomarker for AD because it evaluates the capacity to identify early-stage AD by measuring the plasma Aβ42/40 levels [34,35]. Studies have found that plasma Aβ42/40, combined with age and APOE e4 status, can accurately diagnose cerebral amyloidosis and screen for cerebral amyloidosis in cognitively healthy individuals [36]. Furthermore, individuals with plasma Aβ42/40 positivity have an increased risk of amyloid-beta PET positivity, which may be similar to the correlation between cerebrospinal fluid biomarkers and amyloid-beta PET [37]. Previous studies have indicated that the cerebrospinal fluid Aβ and cerebral amyloidosis levels are associated with individual factors, including genetic type, age, and level of education [38,39,40,41,42]. These factors may also influence the concentration of Aβ in plasma. This study aimed to comprehensively analyze these factors.

In this study, an analysis was conducted on patients with SCD matched for genotype and general clinical data with the control group. Compared with those in the HC group, the patients in the SCD group demonstrated a mild decline in the MoCA scores and a significant increase in the plasma biomarker Aβ42. A data analysis revealed that although the MoCA in the SCD group declined, it was still within the range of normal cognition, and the difference was very subtle. We may need a larger sample size to confirm the existence of this difference further. According to the vascular hypothesis of AD [43], vascular risk factors can lead to the dysfunction of the neurovascular unit (NVU) and tissue hypoxia. Tissue hypoxia may reduce Aβ clearance in the brain, increasing Aβ accumulation in the brain tissue and blood vessels. The increase in Aβ can amplify neuronal dysfunction and neuronal fiber entanglement and accelerate neurodegeneration, leading to dementia. Studies have found that plasma Aβ42 causes damage to the blood vessel wall, leading to microbleeds [44], and Aβ42 exhibits biphasic dynamics [45]. In the pre-pathological stage of Aβ, the plasma Aβ42 concentrations increase and only begin to decrease after the accumulated Aβ42 concentration reaches the threshold for aggregation and plaque formation. This may also be an important reason for the early absence of significant changes in Aβ42/40, which is consistent with our research. Future studies can use multi-center, large-sample, and more accurate plasma Aβ detection methods to verify our results.

The measurement of GMV in the brain has become a mature imaging biomarker for diagnosing cognitive impairment [46]. This study found decreased GMVs in patients in the SCD group in the HIP.L, REC.R, and right PreCG.R, compared with those in the HC group. Reduced GMV values in the hippocampus and entorhinal cortex is a characteristic feature of patients with AD [18]. Furthermore, Peter et al. [47] employed a multivariate pattern analysis to develop a classifier that distinguishes individuals with SCD, amnestic MCI, and cognitively healthy individuals. The voxels that played significant roles in the classification process were mainly located in the hippocampal region and the perirhinal cortex [48]. This is consistent with the results of our study, confirming hypothesis 2. The PreCG.R is part of the frontal lobe and is a higher center for random motion and advanced cognitive activities [49]. Additionally, it is widely connected to the executive function and working memory of the frontal lobe [50]. Buchy et al. found that the PreCG and the frontal lobe cortex in the anterior cingulate cortex play crucial roles in self-referential information processing, including self-concept, self-reflection, and self-awareness [51,52]. Furthermore, Bi et al. [53] identified PreCG as a pathological brain region for late-stage MCI using the genetic evolutionary random forest method. The decline in the volume of the PreCG.R further supports hypothesis 2, indicating that patients with SCD display AD-like structural changes in areas associated with cognition.

The rectus muscle is located on the innermost edge of the ventral surface of the frontal lobe, and its function is not yet clear, as it is sometimes considered a non-functional area in clinical practice. However, Burks et al. reported that the rectus gyrus is a subregion of the orbitofrontal cortex and is closely connected to the memory function network of the frontal lobe [54]. Li et al. [55] demonstrated that asymmetry in the rectus gyrus volume can lead to social function impairments, decreased cognitive empathy, and increased emotional disturbances. Therefore, patients with SCD with atrophy in the precentral and rectus gyrus may experience memory decline and complaints related to confusion in executive function. This is inconsistent with the lack of apparent decline in executive function in the patients with SCD mentioned in this study, suggesting that in the early stages of the disease, those with SCD exhibit more declines in memory.

We further analyzed the correlation between atrophic brain regions, cognitive scales, and plasma biomarkers Aβ42 and Aβ40. Our findings revealed significant positive correlations between the MoCA and GMVs of the HIP.L, REC.R, and PreCG.R, which is consistent with previous findings [8,18]. The decreased GMV in the cognitive functional regions in patients with SCD is closely related to their clinical presentations [56]. Additionally, we found a negative correlation between the right rectus gyrus volume and Aβ42/40 and a negative correlation between the left hippocampal volume and Aβ42. Research has shown that the plasma Aβ may enhance endothelium-dependent vasoconstriction, leading to cerebral hypoperfusion [57,58], and conversely, cerebral hypoperfusion further promotes the excessive production and secretion of Aβ into circulation [59]. This mechanism may be one of the causes of brain atrophy. In the pre-pathological stage of Aβ, the plasma Aβ42 concentrations rise and only begin to decline after the aggregated Aβ42 concentration reaches the threshold for aggregation and plaque formation. Furthermore, this deposition also varies across different brain regions [45]. In the preclinical or prodromal stages of AD, the plasma Aβ40 levels mostly remain unchanged [60]. These two reasons may be important factors leading to the different associations of plasma Aβ subtypes with different brain regions in SCD patients, and we speculate that this association with disease progression and improvement is dynamic.

Additionally, we analyzed changes in the PerAF and brain network topology features in patients with SCD compared to healthy individuals. First, our findings indicate increased PerAF in the left THA SCD, which is consistent with previous reports of low-frequency ALFF and increased PerAF in the bilateral thalamus and the right posterior subregion of the parahippocampal gyrus and bilateral precuneus cortex in patients with SCD. Furthermore, this result supports hypothesis 3. These changes may be early compensatory mechanisms in patients with AD to maintain stable cognitive levels [19,26,61]. However, our study also revealed a negative correlation between PerAF in the left THA and MoCA scores in patients with SCD, suggesting that an increased PerAF is only an early process of compensatory mechanisms for memory processing in the body. The increasing metabolic load in excitatory regions may lead to further neurofunctional decline, resulting in a gradual reduction in the PerAF and the development of MCI and dementia [62,63]. Furthermore, because of the influence of the disease stage, our study observed limited areas with increased PerAF. After adjusting for confounding factors, no significant correlation was found between the PerAF data and plasma biomarkers Aβ42 and Aβ40. This result is partially inconsistent with hypothesis 4. This observation could be attributed to the patients with SCD being in a relatively late stage of the disease.

Second, this study aimed to investigate changes in the gray matter networks in SCD and explore the relationship between plasma Aβ and early brain alterations. Understanding this relationship is crucial to comprehend the pathophysiological basis of AD. In individuals with normal cognition, the brain networks exhibit high-density local connections and a few long-range connections, forming an efficient network with a relatively low wiring cost and optimal adaptability to various environments. This balance between local and global information processing and integration is known as small-world properties [64,65]. This study used brain network modeling to calculate the values for aSigma and aEg, representing small-world properties and global efficiency, respectively. These values were used to investigate the changes in the brain network topology characteristics. The analysis of the topological properties of brain networks in the two groups revealed that the mean aSigma value was lower in the patients in the SCD group than in those in the HC group, indicating a lower small-world property and a tendency of the gray matter network to move toward random organization. Additionally, the values of aEg significantly decreased in the SCD group compared to the control group, indicating a decrease in global effects and implying network disconnection. Our study results suggest that changes in the topological features of gray matter networks can be observed in patients with SCD, which is consistent with the findings of Cantero et al. [20]. This result further supports hypothesis 3.

Ten Kate et al. [66] observed an association between the overall amyloid deposition in patients with SCD, decreased gray matter network clustering, and fewer small-world characteristics. In our study, we found a positive correlation between the aSigma value representing small-world properties and MoCA scores, which is consistent with the findings of Verfaillie et al., indicating that disrupted brain network topological properties are related to cognitive decline and the risk of disease progression [67,68]. However, our results did not indicate a significant correlation between aSigma and plasma Aβ42 or Aβ40. Furthermore, we found that aEg was positively correlated with plasma Aβ42/40. Combining these findings with those from our previous analysis of general clinical data, we speculate that it may be related to a compensatory mechanism of early global efficiency in SCD and the dynamic changes in plasma Aβ [45].

We conducted an analysis of the SCD multimodal MRI and plasma Aβ, identifying early change characteristics that offer a theoretical basis for early intervention. The vascular hypothesis of AD [43] may also provide a scientific basis for exploring changes in SCD multimodal imaging and blood biomarkers through transcranial electrical and magnetic stimulation of cerebral blood flow. This provided us with a broader direction for more in-depth research on SCD in the future. This study had some limitations. First, these results are based on a small sample size; thus, future studies should include larger samples. Second, the analysis focused only on two types of serum amyloid fibrils and did not compare the strengths of the brain network connections. Research on SCD has predominantly centered on the internal connectivity of the DMN, yielding inconsistent results [23,24]. Moving forward, we aim to delve deeper into the role of the DMN network in SCD and the features of other relevant networks. Simultaneously, efforts are underway to classify the origins of SCD, with the findings indicating that clinically diagnosed SCD patients face a greater risk of experiencing imaging changes and developing dementia compared to SCD patients recruited from the community [69]. Therefore, future research with larger sample sizes, a comprehensive analysis of serum markers, and global and local network functional connectivity strength in patients with SCD is needed to validate these findings further.

## 5. Conclusions

In our study, despite some inconsistencies, patients with SCD exhibited GMV reductions and a disruption of gray matter networks, accompanied by varying degrees of changes in Aβ40 and Aβ42. Multiparametric MRI examinations may provide better insights into the pathological changes in individuals with SCD. A previous analysis has shown an increased PerAF in the left THA, suggesting that it is not the primary site of cognitive impairment but a compensatory region in the cognitive system. No single neuroimaging method can accurately depict the full spectrum of pathological mechanisms in SCD. Therefore, the increasing application and maturation of multimodal neuroimaging techniques present an opportunity to explore the complex relationship between amyloid fibrils and cortical functional networks in patients with SCD.

## Figures and Tables

**Figure 1 brainsci-13-01624-f001:**
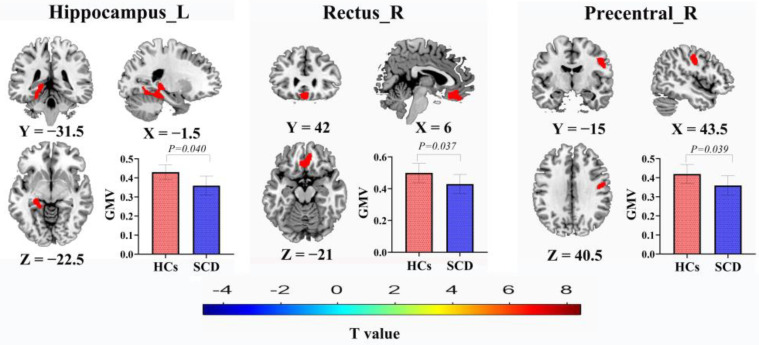
Differences in gray matter volumes between patients in the SCD and HC groups. SCD, subjective cognitive decline; HC, healthy control.

**Figure 2 brainsci-13-01624-f002:**
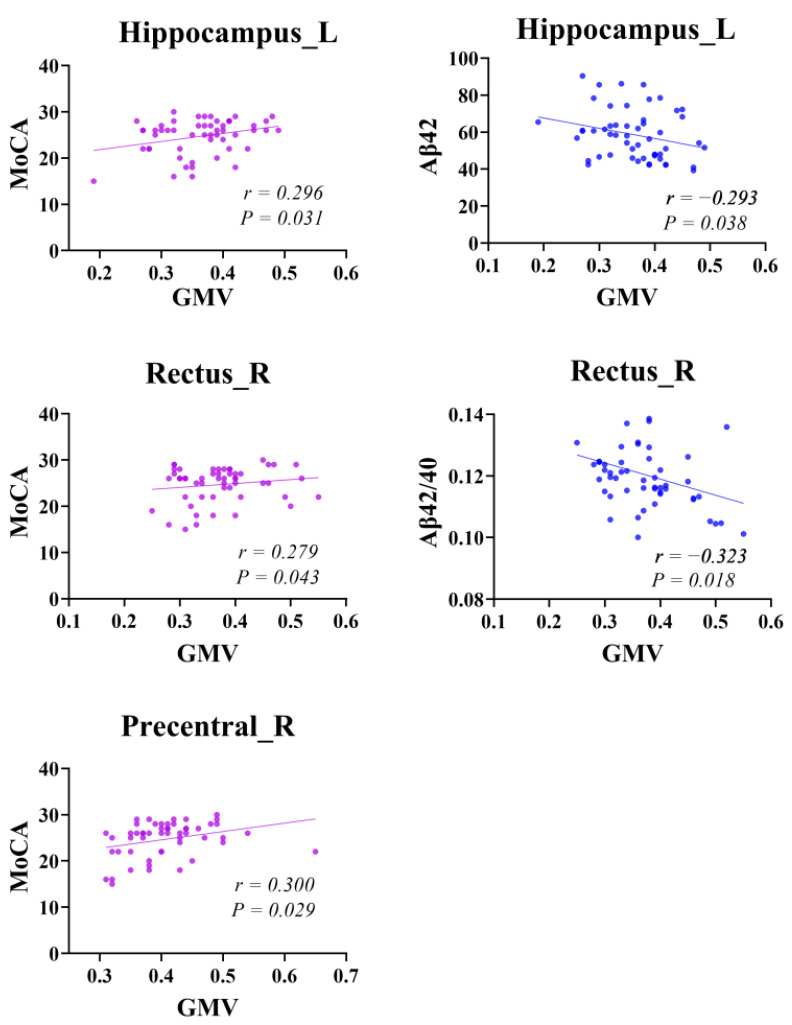
Correlations between GMV of the left hippocampus, right retrosplenial cortex, and right precentral gyrus with MoCA scores, Aβ42, and Aβ42/40. Warm and cool colors indicate positive and negative correlations, respectively. MoCA, Montreal Cognitive Assessment; GMV, gray matter volume.

**Figure 3 brainsci-13-01624-f003:**
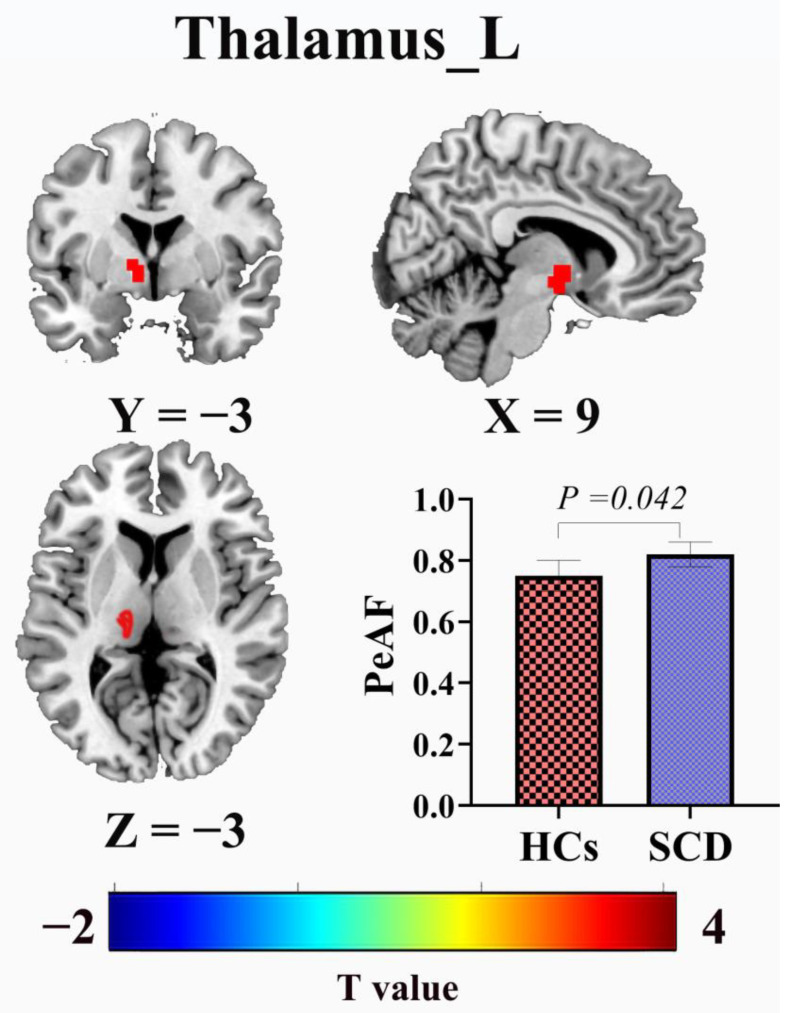
Differences in perirhinal amyloid fibrils (PerAF) between individuals in the SCD and HC groups. SCD, subjective cognitive decline; HC, healthy control.

**Figure 4 brainsci-13-01624-f004:**
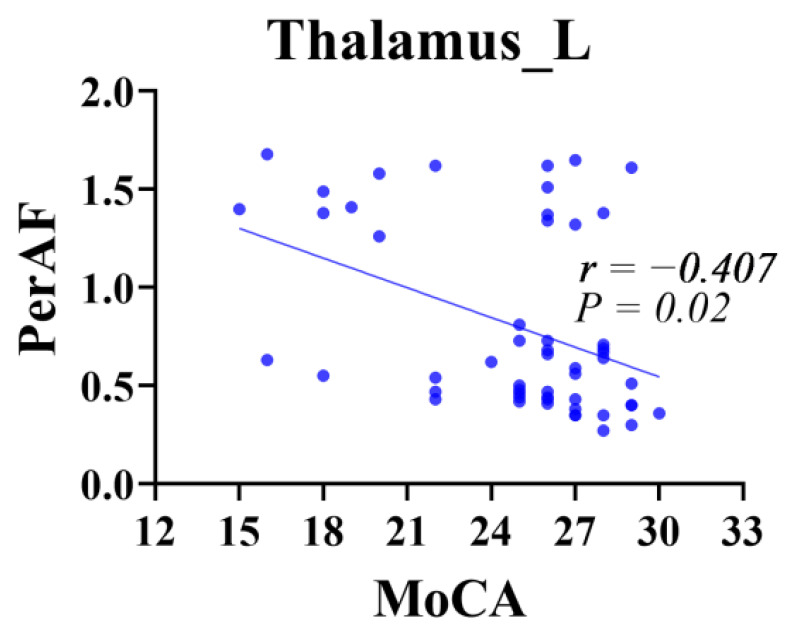
Relationship between PerAF and MoCA scores in individuals with SCD. Cool colors indicate negative correlations. SCD, subjective cognitive decline; MoCA, Montreal Cognitive Assessment.

**Figure 5 brainsci-13-01624-f005:**
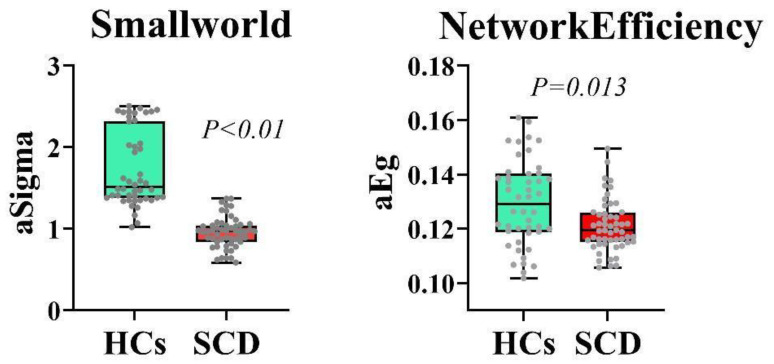
Comparison of the average values of small-world properties and global efficiency between HC and SCD groups’ patients. SCD, subjective cognitive decline; HC, healthy control. aSigma and aEg represent the average values of small-world properties and global efficiency across 19 brain network sparsity levels, respectively.

**Figure 6 brainsci-13-01624-f006:**
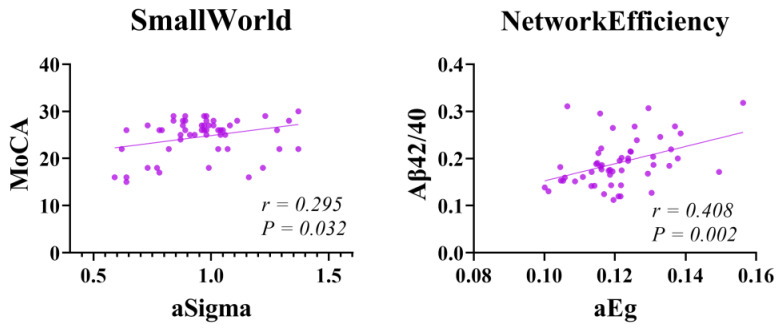
Correlations between brain network topological features and MoCA scores, Aβ42/40. MoCA, Montreal Cognitive Assessment. Warm colors indicate positive correlations.

**Table 1 brainsci-13-01624-t001:** Comparison of clinical characteristics, neuropsychology, and plasma Aβ levels among the two study groups.

	HCs (*n* = 53)	SCD (*n* = 46)	χ^2^/t/U-Value	*p*-Value
Male sex (*n*/%)	31/31.1	36/35.9	0.03 ^a^	0.955
Age (years)	55.85 ± 6.80	55.83 ± 6.49	0.013 ^b^	0.990
Education (years)	8.10 ± 4.05	7.472 ± 3.16	0.283 ^b^	0.754
Hypertension (*n*/%)	6/7.0	9/8.0	0.297 ^a^	0.86
Diabetes (*n*/%)	1/0.9%	1/1.1%	0.010 ^a^	0.919
Hyperlipidemia (*n*/%)	3/4.2%	6/4.8	0.68 ^a^	0.407
Smoking (*n*/%)	6/17.6	14/38.9	5.653 ^a^	0.059
Drinking (*n*/%)	9/26.5	17/47.2	4.350 ^a^	0.114
MMSE (score)	29 (27, 30)	29 (27, 30)	−0.670 ^c^	0.503
MoCA (score)	25 (23, 27)	24 (20, 26)	−1.966 ^c^	0.049
ADL (score)	20 (20, 20)	20 (20, 20)	1.504 ^c^	0.259
CAMCOG (score)	93 (87, 101)	91 (84, 98)	1.313 ^c^	0.294
CDR (score)	0 (0, 0)	0 (0, 0)	−1.871 ^c^	0.061
Aβ42 (pg/mL)	57.50 ± 9.82	61.43 ± 11.13	2.186 ^b^	0.037
Aβ40 (pg/mL)	293.55 ± 54.16	313.25 ± 62.56	1.135 ^b^	0.205
Aβ42/40	0.19 (0.15, 0.24)	0.12 (0.11, 0.12)	−1.024 ^c^	0.306

SCD, subjective cognitive decline; HCs, healthy controls; MMSE, Mini-Mental State Examination; MoCA, Montreal Cognitive Assessment; ADL, Activity of Daily Living Scale; CAMCOG, Cambridge Cognitive Examination-Chinese version; CDR, Clinical Dementia Rating; Aβ, amyloid-beta; Aβ42, amyloid-β42; Aβ40, amyloid-β40; ^a^, the χ^2^ value was tested using the chi-squared test; ^b^, the t value was tested using a two-sample *t*-test; ^c^, the U value was tested using the Mann–Whitney test.

**Table 2 brainsci-13-01624-t002:** Comparison of APOE allele distribution between the two groups (*n*/%).

	HC (*n* = 43)	SCD (*n* = 51)	χ^2^	*p*
E2	14/7.0	17/17.0	0.013	0.993
E3	57/5 6.8	69/69.2		
E4	11/11.3	14/13.7		

APOE, apolipoprotein E; SCD, subjective cognitive decline; HC, healthy control.

**Table 3 brainsci-13-01624-t003:** Comparison of APOE genotype between the two groups (*n*/%).

	HC (*n* = 43)	SCD (*n* = 51)	χ^2^	*p*
E2/E2	3/2.7	3/3.3	2.250	0.134
E2/E3	5/5.3	7/6.7		
E3/E3	22/21.8	27/27.2		
E2/E4	3/3.1	4/3.9		
E3/E4	8/8.0	10/10.0		
E4/E4	0/0	0/0		

APOE, apolipoprotein E; SCD, subjective cognitive decline; HC, healthy control.

## Data Availability

All data generated or analyzed during this study are included in this published article.

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
