# Peer review of "Changes in Multiparametric Magnetic Resonance Imaging and Plasma Amyloid-Beta Protein in Subjective Cognitive Decline"

_brainsci, 2023, doi:10.3390/brainsci13121624_

Round 1
Reviewer 1 Report (Previous Reviewer 1)
Comments and Suggestions for Authors
The study is very interesting and tries to analyze different markers of subjective cognitive decline (SCD). The Introduction covered all aspects of the study, however I do not agree with the term multimodal MRI examinations, as analyzes using different imaging modalities such as MRI, PET, CT, among others, were not performed. The most correct would be "multiparametric MRI examinations" used in line 75. In item 2.2 of materials and method, the inclusion of 47 HC was described, but in the abstract it is 46, which is correct?
The method is very well detailed and I have no suggestions to give. In the results session, the authors described the results very well, I would only suggest the use of boxplot with jitter in figure 6, instead of violin form. The discussion and conclusion are appropriate.
Author Response
Question 1. however I do not agree with the term multimodal MRI examinations, as analyzes using different imaging modalities such as MRI, PET, CT, among others, were not performed. The most correct would be "multiparametric MRI examinations" used in line 75. In item 2.2 of materials and method, the inclusion of 47 HC was described, but in the abstract it is 46, which is correct?
Reply: We appreciate the suggestion from the reviewer. We acknowledge that the expression and writing were not precise, and we have verified the data accordingly. We have now made the necessary corrections.
Question 2. suggest the use of boxplot with jitter in figure 6, instead of violin form.
Reply: We appreciate the suggestion from the reviewer. Using a boxplot with jitter makes the comparison clearer and more visually appealing, and we have made the necessary corrections.

Reviewer 2 Report (Previous Reviewer 3)
Comments and Suggestions for Authors
This paper titled ‘Changes in Multimodal Imaging and Plasma Amyloid-beta Protein in Subjective Cognitive Decline’ investigates Subjective Cognitive Decline (SCD) and its potential association with neural biomarkers, focusing on plasma amyloid-beta (Aβ) levels and structural brain changes. The study compares 53 patients with SCD and 46 matched controls and analyzes demographic, clinical, and neuroimaging data.
The study reveals a mild decline in Montreal Cognitive Assessment (MoCA) scores among the SCD group compared to healthy controls (HC). Additionally, significantly higher plasma Aβ40 levels are observed in the SCD group. Logistic regression analysis confirms the statistical significance of these differences, demonstrating that decreased MoCA scores and increased Aβ40 levels are independent risk factors for SCD.
Structural MRI analysis uncovers reduced gray matter volume (GMV) in specific brain regions of patients with SCD, including the left hippocampus (HIP.L), right rectal gyrus (REC.R), and right precentral gyrus (PreCG.R). Correlation analysis reveals associations between MoCA scores and GMV in these regions. Furthermore, Aβ40 and Aβ42/40 levels show negative correlations with specific brain volumes, suggesting associations between increased plasma amyloid levels and structural brain changes.
The paper emphasizes the potential of plasma Aβ40/42 ratios as biomarkers for early-stage Alzheimer's disease (AD). It discusses the implications of the observed GMV reductions in the context of cognitive impairment and indicates that structural alterations in regions associated with cognitive function support the hypothesis of SCD-related changes resembling AD-like structural patterns.
The research underscores the importance of multimodal neuroimaging techniques in understanding the complex relationships between plasma amyloid fibrils and cortical functional networks in patients with SCD. Despite some inconsistencies, the study shows promise in detecting GMV reductions and gray matter network disruptions in patients with SCD, alongside fluctuations in Aβ40 and Aβ42.
In general, I think the idea of this article is really interesting and the authors’ fascinating observations on this timely topic may be of interest to the readers of Brain Sciences. However, some comments, as well as some crucial evidence that should be included to support the author’s argumentation, needed to be addressed to improve the quality of the manuscript, its adequacy, and its readability prior to the publication in the present form. My overall judgment is to publish this paper after the authors have carefully considered my suggestions below, in particular reshaping parts of the ‘Introduction’ and ‘Methods’ sections by adding more evidence.
Please consider the following comments:
• The introduction effectively sets the stage and provides a comprehensive overview of the existing literature on SCD, presenting the knowledge gaps and the need for the study. However, the structure could benefit from clearer demarcation of sections. The information seems densely packed, making it challenging for readers to follow each point's progression.
• The study's methodology is well-documented, covering ethical considerations, participant selection criteria, neuroimaging methods, and plasma biomarker detection. However, there's an extensive description that might benefit from a more concise presentation, focusing on key details without overwhelming the reader. In this regard, to further enhance the understanding of the neural substrates associated with SCD, it might be beneficial delving into the specific neural networks implicated in SCD could be enlightening. Detailing the involvement of default mode network (DMN), salience network, and other relevant networks could shed light on how alterations in these networks might relate to the cognitive decline observed in SCD [1-4].
• The approach for data preprocessing, network analysis, and statistical analyses is clear, but certain technical descriptions might be too detailed for readers not well-versed in these methods.
• Data Analysis and Results: The presentation of results seems to be missing from this excerpt. It would be essential to include the findings to judge the study's ability to address the hypotheses and contribute to the existing knowledge.
• Statistical Interpretation: While the statistical analysis is described, some interpretation of the clinical relevance of the findings might enhance the section. For instance, discussing the practical implications of certain values or associations could make the manuscript more insightful.
• Discussion: The discussion does well to link the findings to previous research. However, it would be beneficial to widen the context, perhaps by discussing the limitations of the study in greater depth and exploring how the findings contribute to the broader field of study.
• Future Directions: Adding suggestions for further research or future directions could improve the completeness of the manuscript.
• Limitations and Future Work: The limitations mentioned in the manuscript could be expanded upon to provide a more comprehensive understanding of the constraints and scope of the study. Furthermore, the manuscript briefly touches on suggestions for future studies, but a more detailed exploration could enhance the conclusions drawn from this research.
• References: Authors should consider revising the bibliography, as there are several incorrect citations. Indeed, according to the Journal’s guidelines, they should provide the abbreviated journal name in italics, the year of publication in bold, the volume number in italics for all the references. Also, please correct in-text citations: reference should be numbered, and placed in square brackets [ ] (for example [1]).
• I would suggest discussing the practical implications of the study’s findings in clinical practice could be a valuable addition, as well emphasizing the potential clinical utility of the findings could be helpful in drawing the interest of clinicians and researchers.
References:
1. https://doi.org/10.1111/acps.13602
2. https://doi.org/10.3390/ijms242115739
3. DOI: 10.1111/psyp.14020
Comments on the Quality of English Language
Minor English check is required.
Author Response
Responses to the Reviewers’ comments:
Question 1. The introduction effectively sets the stage and provides a comprehensive overview of the existing literature on SCD, presenting the knowledge gaps and the need for the study. However, the structure could benefit from clearer demarcation of sections. The information seems densely packed, making it challenging for readers to follow each point's progression.In this regard, to further enhance the understanding of the neural substrates associated with SCD, it might be beneficial delving into the specific neural networks implicated in SCD could be enlightening. Detailing the involvement of default mode network (DMN), salience network, and other relevant networks could shed light on how alterations in these networks might relate to the cognitive decline observed in SCD
Reply: Thank you for your suggestion. Accordingly, we have restructured the introduction section, with the first paragraph introducing the concept of SCD and the second and third paragraphs elaborating on SCD research using various MRI modalities, such as DTI, structural MRI, and resting-state fMRI, and reporting on DMN research in SCD. The third paragraph briefly outlines the current research status of plasma Aβ in SCD. Finally, four hypotheses regarding the relationship between multimodal MRI, plasma Aβ, and neuropsychological scales are proposed. Thank you again; we believe this has improved the organization of the article.
Question 2. The approach for data preprocessing, network analysis, and statistical analyses is clear, but certain technical descriptions might be too detailed for readers not well-versed in these methods.
Reply: Thank you for your comment. We have trimmed down the introduction section while ensuring the replicability of the research, making the article more concise, per your suggestion.
Question 3. Data Analysis and Results: The presentation of results seems to be missing from this excerpt.
Reply: Thank you for this suggestion. We agree that a description on this was lacking. We have now supplemented the results section to make the article more complete while also explaining the significance of the study.
Question 4. Statistical Interpretation: While the statistical analysis is described, some interpretation of the clinical relevance of the findings might enhance the section. For instance, discussing the practical implications of certain values or associations could make the manuscript more insightful.
Reply: Thank you for your suggestion. We have reorganized the results section to provide clearer explanations and further elaborated on the results in the discussion section accordingly.
Question 5. Discussion: The discussion does well to link the findings to previous research. However, it would be beneficial to widen the context, perhaps by discussing the limitations of the study in greater depth and exploring how the findings contribute to the broader field of study.
Reply: Per your suggestion, we have restructured the discussion section, analyzing our proposed four hypotheses and our conclusions. We compared our findings with previous research, emphasizing similarities and differences, and identified limitations in our study. This has enhanced the completeness of the article and we thank you for this suggestion.
Question 6. Further elaboration on future work, research directions, and limitations of the article.
Reply: Thank you for this suggestion. We conducted an analysis of SCD multimodal MRI and plasma Aβ, identifying early change characteristics that offer a theoretical basis for early intervention. The vascular hypothesis of AD may also provide a scientific basis for exploring changes in SCD multimodal imaging and blood biomarkers through transcranial electrical and magnetic stimulation on cerebral blood flow. This has provided us with a broader direction for more in-depth research on SCD in the future.
This study had some limitations. First, these results are based on a small sample size; thus, future studies should include larger samples. Second, the analysis focused only on two types of serum amyloid fibrils and did not compare the strength of brain network connections. The current research on SCD predominantly centers on variations within the intrinsic functional network of the DMN, yielding inconsistent findings. Moving forward, we aim to delve deeper into the role of the DMN network in SCD and the features of other relevant networks. Simultaneously, efforts are underway to classify the origins of SCD, with findings indicating that clinically diagnosed SCD patients face a greater risk of experiencing imaging changes and developing dementia compared to SCD patients recruited from the community [69]. Therefore, future research with larger sample sizes, a comprehensive analysis of serum markers, and global and local network functional connectivity strength in SCD patients is needed to further validate these findings.
We have included this information in the manuscript.
Question 7.References: Authors should consider revising the bibliography, as there are several incorrect citations. Indeed, according to the Journal’s guidelines, they should provide the abbreviated journal name in italics, the year of publication in bold, the volume number in italics for all the references. Also, please correct in-text citations: reference should be numbered, and placed in square brackets [ ] (for example [1]).
Reply: Thank you for pointing this out, we apologize for the errors in this regard. We thoroughly reviewed the references and made corrections as necessary.
Question 7.suggest discussing the practical implications of the study’s findings in clinical practice could be a valuable addition, as well emphasizing the potential clinical utility of the findings could be helpful in drawing the interest of clinicians and researchers.
Reply: Thank you for this suggestion. We have discussed the significance of the limitations section to the research. We analyzed SCD multimodal MRI and plasma Aβ, identifying early change characteristics that offer a theoretical basis for early intervention. The vascular hypothesis of AD may also provide a scientific basis for exploring changes in SCD multimodal imaging and blood biomarkers through transcranial electrical and magnetic stimulation on cerebral blood flow. This has provided a broader direction for more in-depth research on SCD in the future.

Reviewer 3 Report (Previous Reviewer 2)
Comments and Suggestions for Authors
Dear Authors, I read with interest your revised manuscript, now entitled “Changes in Multimodal Imaging and Plasma Amyloid-beta Protein in Subjective Cognitive Decline”. The manuscript has been well implemented and is very interesting, with a particular detailed description and interpretation of imaging findings. However, some points deserve to be further improved. I report below some comments and suggestions.
1) The following sentence in the Discussion should be changed. “Currently, the assessment and prediction of cognitive function using cerebrospinal fluid biomarkers and amyloid positron emission tomography (PET) are not only invasive but also expensive, with potential gamma radiation exposure for both doctors and particiipants.” At first, the use of these methods is not dedicated to the assessment and prediction of cognitive functioning, as well assessed by functional MRI or other techniques used in cognitive neuroscience. Amyloid PET and CSF biomarkers are diagnostic markers for an accurate diagnosis of Alzheimer’s disease, or to rule out some pathological manifestations. Also, the limitations of the two methods are different, for example the radiactivity does not involve the lumbar puncture procedure. Please modify the paragraphs.
2) The authors deal with Aβ, Aβ40 and Aβ42/40 ratio in different passages. Please specificy the clinical or research significance of results taken into account the different pathopshyisological meaning of these distinct isoforms. For example, the association between MoCA and Aβ40 levels is different than the use of a Aβ40/42 ratio. Also, the sentence “This further emphasizes the underlying causes of cognitive impairment in patients with SCD.” is misleading, since SCD by definition does not encompass cognitive impairment. A certain degree of cognitive dysfunction can be present, in a subtle magnitude. This is interesting, but it needs to be better argumented. The difference in MoCA scores between SCD and HC groups might be subtle, rather than mild. I strongly suggest to further explain the previous findings about decreased MoCA scores and increased Aβ40 levels as independent risk factors for SCD.
Also, the following sentence deserves a further clarification. “Additionally, we found a negative correlation between left hippocampal volume and Aβ40 and a negative correlation between right rectus gyrus volume and Aβ42/40. These findings indicate that changes in GMV in patients with SCD are associated with increased plasma amyloid beta levels, consistent with the pathological changes of cognitive decline in AD [56].” The correlations described above are very specific, and observed with Aβ40 only or with Aβ42/40 ratio. I wander that this distinct profile may have distinct pathophysiological mechanisms. Also, the final sentence is too wide in relation to the specific findings. Please modify by specifying the type of Aβ isoform and the possible implications of these findings with respect to previous ones.
3) In the methods, author report that the cognitive evaluation has been carried out by means of a neuropsychological assessment including MMSE and MoCA. Were other neuropsychological tests administered to subjects? The only use of MMSE and MoCA is not recommeded to classify a subject as SCD, because a in-depth assessment with domain-specific tests is mandatory to rule out potential mild cognitive impairment. The activities of daily living are less informative, for the same reason. At the same time, the lack of diagnostic markers for AD does not allow to consider these subjects in the AD continuum in the sense of ATN framework. I suggest to implement the limits accordingly.
4) Study design. Is there a methodological reason why left-handed individuals are excluded? I believe that Citation 7 (Jessen et al.) is misleading in the section Methods. In fact, the prospect of SCD-plus is different by the inclusion criteria in the present study.
Comments on the Quality of English Language
Not relevant.
Author Response
Responses to the Reviewers’ comments:
Question 1. Amyloid PET and CSF biomarkers are diagnostic markers for an accurate diagnosis of Alzheimer’s disease, or to rule out some pathological manifestations. Also, the limitations of the two methods are different, for example the radiactivity does not involve the lumbar puncture procedure. Please modify the paragraphs.
Reply: Thank you for pointing this out. We apologize for the lack of clarity in this section. This section has been corrected as follows:
Cerebrospinal fluid (CSF) Ab42/40 (or CSF Ab42) and Ab-PET are biomarkers used to detect Ab pathology in the brain and are considered early indicators of cognitive impairment [32]. However, the collection of CSF through lumbar puncture is often viewed as a more invasive diagnostic method, while Ab-PET scans can be costly and pose potential risks of gamma radiation exposure for both medical professionals and patients.
Question 2. This specifically explains the pathological and research significance of different subtypes of Ab, while also addressing the previous findings of decreased MoCA scores and increased Aβ40 levels as independent risk factors for SCD.
Reply: Thank you for your comment. We reanalyzed the data and provided a more detailed explanation in the conclusion section as below. This makes the article more comprehensive.
In this study, analysis was conducted on patients with SCD matched for genotype and general clinical data with the control group. Compared with those in the HC group, the patients in the SCD group demonstrated a mild decline in MoCA scores and a significant increase in the plasma biomarker Aβ42. Data analysis revealed that although MoCA in SCD had declined, it was still within the range of normal cognition, and the difference was very subtle. We may need a larger sample size to confirm the existence of this difference further. According to the vascular hypothesis of AD [43], vascular risk factors can lead to the dysfunction of the neurovascular unit (NVU) and tissue hypoxia. Tissue hypoxia may reduce Aβ clearance in the brain, increasing Aβ accumulation in brain tissue and blood vessels. The increase in Aβ can amplify neuronal dysfunction and neuronal fiber entanglement and accelerate neurodegeneration, leading to dementia. Studies have found that plasma Aβ42 causes damage to the blood vessel wall, leading to microbleeds [44], and Aβ42 exhibits biphasic dynamics [45]. In the pre-pathological stage of Aβ, plasma Aβ42 concentrations increase and only begin to decrease after the accumulated Aβ42 concentration reaches the threshold for aggregation and plaque formation. This may also be an important reason for the early absence of significant changes in Aβ42/40, which is consistent with our research. Future studies can use multi-center, large-sample, and more accurate plasma Aβ detection methods to verify our results.
Question 3.the following sentence deserves a further clarification. “Additionally, we found a negative correlation between left hippocampal volume and Aβ40 and a negative correlation between right rectus gyrus volume and Aβ42/40. These findings indicate that changes in GMV in patients with SCD are associated with increased plasma amyloid beta levels, consistent with the pathological changes of cognitive decline in AD [56].” The correlations described above are very specific, and observed with Aβ40 only or with Aβ42/40 ratio. I wander that this distinct profile may have distinct pathophysiological mechanisms. Also, the final sentence is too wide in relation to the specific findings. Please modify by specifying the type of Aβ isoform and the possible implications of these findings with respect to previous ones.
Reply: Thank you for carefully evaluating our manuscript and providing this suggestion. We have modified the conclusion accordingly, as below:
We further analyzed the correlation between atrophic brain regions, cognitive scales, and plasma biomarkers Aβ42 and Aβ40. Our findings revealed significant positive correlations between MoCA and GMVs of the HIP.L, REC.R, and PreCG.R, consistent with previous findings [18, 8]. The decreased GMV in cognitive functional regions in patients with SCD is closely related to their clinical presentations [56]. Additionally, we found a negative correlation between the right rectus gyrus volume and Aβ42/40 and a negative correlation between the left hippocampal volume and Aβ42. Research has shown that plasma Aβ may enhance endothelium-dependent vasoconstriction, leading to cerebral hypoperfusion [57,58], and conversely, cerebral hypoperfusion further promotes excessive production and secretion of Aβ into circulation [59]. This mechanism may be one of the causes of brain atrophy. In the pre-pathological stage of Aβ, plasma Aβ42 concentrations rise and only begin to decline after the aggregated Aβ42 concentration reaches the threshold for aggregation and plaque formation. Furthermore, this deposition also varies across different brain regions [45]. In the preclinical or prodromal stages of AD, plasma Aβ40 levels mostly remain unchanged [60]. These two reasons may be important factors leading to the different associations of plasma Aβ subtypes with different brain regions in SCD patients, and we speculate that this association with disease progression and improvement is dynamic.
Question 4. In the methods, author report that the cognitive evaluation has been carried out by means of a neuropsychological assessment including MMSE and MoCA. Were other neuropsychological tests administered to subjects? The only use of MMSE and MoCA is not recommeded to classify a subject as SCD, because a in-depth assessment with domain-specific tests is mandatory to rule out potential mild cognitive impairment. The activities of daily living are less informative, for the same reason. At the same time, the lack of diagnostic markers for AD does not allow to consider these subjects in the AD continuum in the sense of ATN framework. I suggest to implement the limits accordingly.
Reply: Thank you for your constructive comment. We acknowledge this error and have included MMSE, MoCA, ADL, and Cambridge Cognitive Examination-Chinese version (CAMCOG-C) scores in the inclusion criteria. We have also restated the inclusion criteria for clarity.
Question 5.Study design. Is there a methodological reason why left-handed individuals are excluded? I believe that Citation 7 (Jessen et al.) is misleading in the section Methods. In fact, the prospect of SCD-plus is different by the inclusion criteria in the present study.
Reply: Thank you for your evaluation and suggestion. We have included a supplementary explanation to the inclusion criteriaas follows:
We included 53 patients with SCD and 46 age- and sex-matched HCs. The inclusion criteria for the SCD group were as follows: (1) age between 55 and 75 years; (2) SCD reported; and (3) cognitive psychological assessment results as follows: MoCA scores >19, >22, and >24 for elementary school education or below, middle school education, and university education, respectively; Mini-Mental State Examination (MMSE) scores >17, >20, and >24 for illiterate individuals, elementary school education, and junior high school education or higher, respectively; Cambridge Cognitive Examination-Chinese version (CAMCOG-C) scores ≥90 and Clinical Dementia Rating (CDR) score of 0. The exclusion criteria were as follows: (1) severe cerebrovascular disease; (2) other diseases or factors that may cause cognitive decline, such as physical illnesses, immune abnormalities, thyroid dysfunction, depression, history of brain injury, psychiatric history, drug dependence, and alcohol intoxication; (3) left-handedness; and (4) inability to cooperate with MRI scanning, plasma collection, and cognitive testing. The inclusion criteria for the HC group were as follows: (1) no evidence of cognitive impairment and (2) no brain atrophy or apparent white matter hyperintensities observed on cranial imaging.

Round 2
Reviewer 3 Report (Previous Reviewer 2)
Comments and Suggestions for Authors
Dear authors, the manuscript sounds well written, it is elaborate and well revised. I have a concern about a methodological point regarding ADL and IADL, that should be explained.
1. Methods. Study design and participants.
It's not clear the guidelines used for ADL and IADL scoring. In particular, the following sentence: "score >16 indicates varying degrees of functional decline; if two or more items score ≥3 points or the total score is ≥22, it indicates significant functional impairment." should be argumented according to the scoring system used. Since ADL score range is 0-6 and IADL score range is 0-8, with higher scores indicative of better functional status, it appears less clear the system used.
There are also minor points to change.
2. Methods. Study design and participants.
Please replace "cognitive psychological assessment" with "cognitive assessment" or "neuropsychological assessment".
3. Methods. Study design and participants.
It's still not clear the reason of exclusion for left-handedness individuals.
4. Discussion. I suggest to modify and attenuate the sentence stating that CSF and amyloid PET are early indicators of cognitive impairment. Rather, in a broader sense, thay may work as early biomarkers related to cognitive dysfunctions.
Author Response
Question 1. Methods. Study design and participants.
It's not clear the guidelines used for ADL and IADL scoring. In particular, the following sentence: "score >16 indicates varying degrees of functional decline; if two or more items score ≥3 points or the total score is ≥22, it indicates significant functional impairment." should be argumented according to the scoring system used. Since ADL score range is 0-6 and IADL score range is 0-8, with higher scores indicative of better functional status, it appears less clear the system used.
Reply: We appreciate the suggestion from the reviewer. We acknowledge that the expression and writing were not precise, and we have verified the data accordingly. We have now made the necessary corrections. The ADL scale, which has been revised by Professor Zhang Mingyuan, is a Chinese version consisting of 20 items that are answered by the subject or an informed person. A total score of 20 indicates complete normality, while a score greater than 20 suggests varying degrees of functional decline, with 80 being the highest achievable score. Individual ability analysis is divided into two levels: a score of 1 indicates normality, while scores of 2-4 indicate functional decline.
Question 2. Methods. Study design and participants. Please replace "cognitive psychological assessment" with "cognitive assessment" or "neuropsychological assessment".
Reply: We appreciate the suggestion from the reviewer. In our paper, we replaced cognitive psychological assessment with neuropsychological assessment.
Question 3. Methods. Study design and participants.It's still not clear the reason of exclusion for left-handedness individuals.
Reply: We appreciate the suggestion from the reviewer. We exclude left-handed individuals from the participant pool, as their brain organization and cognitive processing may differ from right-handed individuals, potentially confounding the study results. This is done to ensure homogeneity among the study participants.
Question 4. Discussion. I suggest to modify and attenuate the sentence stating that CSF and amyloid PET are early indicators of cognitive impairment. Rather, in a broader sense, thay may work as early biomarkers related to cognitive dysfunctions.
Reply: We appreciate the suggestion from the reviewer. Our statement was too absolute, and we have now modified the wording.Cerebrospinal fluid (CSF)Aβ42/40 (or CSF Aβ42) and Aβ-PET are commonly used as biomarkers to detect Aβ pathology in the brain and may work as early biomarkers related to cognitive dysfunctions.[32].

This manuscript is a resubmission of an earlier submission. The following is a list of the peer review reports and author responses from that submission.
Round 1
Reviewer 1 Report
Comments and Suggestions for Authors
The theme of the article is very interesting and has a current relevance for the clinical and scientific community with the increase in the longevity of the population and the greater investment in early predictors of cognitive decline. The introduction addressed all aspects of the article and highlighted a new proposal in the functional analysis of the brain in the resting state, however the methodological description of this new proposal was a little superficial, not making it clear what would change from what has already been done in the analysis. of functional. Only general aspects of functional analysis were described in the methodology. In the morphometric analysis, the methodological detail used was also very simplistic, even the issues of normalization by intracranial volume were reported. Important details for the replicability of the study, since another analysis proposal is being suggested. The description of the result could also be a little more rigorous, showing whether the significant demographic characteristics, when adjusted to the analysis, still maintain the significant result, in order to have a better interpretation of the influence of each factor. The application of these suggestions can greatly improve the methodological clarity and interpretation of the results, since it is a new proposal that is being discussed.
Comments on the Quality of English LanguageAdequate
Author Response
Question 1. Regarding the criticism that the brain network topology analysis method is relatively simple and lacks a deeper analysis of brain network characteristics
Reply: We appreciate the reviewer's suggestion. Currently, there is a lack of consensus in the literature regarding the findings of multimodal magnetic resonance imaging in patients with SCD. In this study, we focused on small-world properties, node clustering coefficients and global efficiency of the topological features in SCD patients. We also used PerAF instead of ALFF and fALFF to analyze the resting-state fMRI low-frequency signals, as PerAF is more reliable and sensitive than ALFF and fALFF. However, we did not analyze the functional connectivity strength and independent components in the brain network characteristics, which provides valuable insights for our future research. In future studies, we plan to conduct more extensive analyses with larger sample sizes and investigate the changes in connectivity strength between different regions of the brain network.
Question 2. Regarding the comment that the description of the voxel-based morphological analysis process is relatively simple,
Reply: We appreciate the reviewer's suggestion. To enhance the study's reproducibility, we have provided detailed supplementary information on the preprocessing of all participants' MRI scans. Additionally, we have described the evaluation of image quality.
The CAT12 toolbox, which is based on Statistical Parametric Mapping 12 (SPM12, see text footnote 1), was utilized to conduct voxel-based morphometry analysis. The general process is as follows:
Segmentation: T1-weighted structural images are segmented into gray matter, white matter, cerebrospinal fluid, and other tissues using tissue probability maps (TPMs); Normalization: The segmented brain tissues are spatially normalized; Jacobian modulation: The gray matter volume is modulated using the Jacobian determinant generated during spatial normalization; Smoothing: The gray matter volume density map modulated by Jacobian is further smoothed to reduce spatial noise. The smoothing kernel (FWHM, full width half maximum) was selected as 6*6*6 mm.
First, the image quality of each subject is manually checked, and those who do not meet the standards are excluded, including obvious signal loss, artifacts, head motion artifacts, and other apparent image issues. The weighted average image quality rating (IQR) is used as the criterion for evaluating image quality, and subjects with a weighted average (IQR) < 70% are excluded.
Question 3. Regarding the suggestion to include demographic characteristics in the results section to enhance rigor
Reply: We appreciate the reviewer's insight. When comparing the two groups, we conducted statistical analyses with demographic covariates such as gender, age, total intracranial volume (TIV), and education level.

Reviewer 2 Report
Comments and Suggestions for Authors
Dear Authors, I read with very interest your manuscript entitled “Changes in Blood Biomarkers and Cortical Network Topological Features in the Preclinical Stage of Alzheimer's Disease: A Multimodal Imaging Analysis”. I report below some comments and suggestions.
The technical section is very well explained. The data about imaging analyses and the argumentation about the main findings are very specific and interesting. However, from a methodological point of view, some points should be improved.
1) The study is structured as an explorative study on the association between blood, imaging and neuropsychological data on a cohort of SCD patients. Although some studies included the SCD in the AD continuum and documented an higher risk of decline in people with SCD, it’s not methodologically correct to define these patients as affected by preclinical Alzheimer’s disease. In the text, these subjects are presented as preclinical AD. However, in absence of a diagnostic characterization with biomarkers, as stated by the ATN Framework, SCD is not classifiable as pre-AD. In the manuscript, the terms “SCD” and “preclinical AD” are used interchangeable. This is not correct, so I recommend to underline this aspect in the limits and to change the nomenclature in the text (methods, results etc.). This represents the main aspect to ameliorate.
2) The main data seems to regard the imaging analyses. So, I suggest to highlight these results as primary finding, and to change title, background and discussion accordingly. In the aims in the Introduction, the work hypothesis is that blood biomarkers are higher in SCD than controls. However, only amyloid has been measured. I recommend to further explain the research purpose to only analyze β-amyloid, among biomarkers; I also recommend to take into account the difference in diagnostic performance between blood vs. cerebrospinal fluid measurement of amyloid in the first section of Discussion, by specifying what previous studies used CSF and what blood among those mentioned.
3) The section about the role of REC.R and PreCG.R is interesting. However, no difference occurred between groups in activities of daily living, as expected, so it’s not clear in what sense the association of executive functions with ADLs is applicable in this study. Please further explain. Also, SCD clinical presentation seems to be mainly in a “dysexecutive” form, if I well understand. This finding partially contrasts with the SCD-I criteria in which memory decline is the most important feature for a standardized definition of the entity (i.e., AD-like associated features of SCD).
4) In the Introduction, it’s correctly mentioned the help seeking as an important aspect in SCD. It could be interesting to further discuss the different patterns of structural imaging between community vs clinical cohort of SCD patients (Perrotin et al., 2017; Pini, Wennberg, 2021).
5) Study design. Is there a methodological reason why left-handed individuals are excluded? I believe that Citation 7 (Jessen et al.) is misleading in the section Methods. In fact, the prospect of SCD-plus is different by the inclusion criteria in the present study.
6) There are some limits that should be further discussed. The use of MMSE and MoCA may reduce the range of findings since the SCD group is by definition a cohort with no cognitive impairment, and very subtle decline can be capture only by more in-depth cognitive assessment. The activities of daily living are less informative, for the same reason. The term “cognitive psychological assessment” should be changed. I suggest to replace with "neuropsychological scales/assessment". The lack of diagnostic markers for AD does not allow to consider these subjects in the AD continuum in the sense of ATN framework.
7) I suggest to change the following points. Pag. 1 line 42-43: to be moved before line 53; Discussion, pag. 12, line 349: add citation; Line 361-368; line 369-380; line 381-392: I suggest to further match all these sections with previous ones in which the same shared argument is dealt with.
Comments on the Quality of English LanguageIn general, the text is well written, little improvement is suggested.
Author Response
We thank you for your thoughtful suggestions and insights. The manuscript has benefited from these insightful suggestions.
The manuscript has been rechecked and the necessary changes have been made in accordance with the your suggestions. Changes are denoted in red text.
Question 1. Defining SCD patients as a methodological approach to preclinical Alzheimer's disease is incorrect.
Reply: We thank the reviewer for the suggestion. Indeed, directly labeling such patients as preclinical AD is not accurate. We have made the necessary corrections in the paper to address this issue.
Question 2. Relevance of results to the article title.recommend to take into account the difference in diagnostic performance between blood vs. cerebrospinal fluid measurement of amyloid in the first section of Discussion
Reply: We thank the reviewer for the suggestion. In this paper, we mainly focused on the relationship between plasma amyloid-beta protein, neuropsychological scales, and brain volume networks. We have emphasized the article's main theme in the title, background, and discussion sections, as per the reviewer’s advice. Additionally, as suggested, in the discussion section, we have compared the advantages , disadvantages of plasma and cerebrospinal fluid detection of amyloid-beta.
Question 3. Unclear explanation of the impact of the REC.R and PreCG.R regions on working memory and executive function.
Reply: We thank the reviewer for the comment. Regarding the inconsistency with the ADL scale, we have rephrased it in the manuscript as follows: SCD patients with atrophy in the REC.R and PreCG.R regions may experience memory decline and complaints related to executive function disturbances, which is inconsistent with the absence of obvious decline in executive function in this study. It is possible that in the early stages of the disease, the predominant manifestation is memory decline,and this will be explained in the paper. In our future work, we will follow up with SCD patients with atrophy in the posterior cingulate and anterior cingulate regions to document changes in their executive function.
Question 4. Potential differences in structural imaging between SCD patients from the community and clinical cohorts.
Reply: We thank the reviewer for the suggestion, which provides new insights for our future research. This study only included patients who reported memory decline clinically and not SCD patients from the community. In future studies, we will compare blood markers such as amyloid-beta and imaging findings in these patients.
Question 5. Exclusion of left-handed patients.
Reply: We thank the reviewer for pointing out the oversight in the literature citation. The inclusion criteria of this study were based on SCD-plus, but with some differences. The participants in this study were right-handed individuals. Due to the scarcity of left-handed individuals during the participant selection process, we opted for a larger number of right-handed individuals to prevent inconsistencies related to dominant hemispheres.
Question 6. Change "cognitive psychological assessment" to "neuropsychological scales/assessment."
Reply:We thank the reviewer for the suggestion.This study found that SCD patients had a mild decline in MoCA compared to HC, but it still fell within the normal range.We agree with the reviewer's suggestion as it makes the article more precise.
Question 7. Consolidate the analysis of the same argument in one discussion section.
Reply: We would like to express our gratitude to the reviewer for their valuable suggestion. In response, we have restructured the discussion section to present the arguments in a more logical and cohesive manner. Furthermore, we have incorporated relevant references in the appropriate sections.

Reviewer 3 Report
Comments and Suggestions for Authors
The paper by Xu and colleagues entitled ‘Changes in Blood Biomarkers and Cortical Network Topological Features in the Preclinical Stage of Alzheimer's Disease: A Multimodal Imaging Analysis’ focuses on a condition known as Subjective Cognitive Decline (SCD), where individuals who still have normal cognitive abilities express concerns about memory problems. Researchers have identified SCD as an early stage of Alzheimer's disease (AD), occurring well before the more recognizable signs of mild cognitive impairment (MCI). One of the primary challenges with SCD is diagnosing it accurately and understanding the underlying mechanisms. People with SCD tend to perform well on cognitive tests but report subjective cognitive decline, making it difficult to pinpoint their condition. Moreover, research on SCD has produced inconsistent results, especially concerning changes in brain volume and blood biomarkers like amyloid proteins. To address these challenges, some researchers have proposed diagnostic criteria for SCD related to AD. These criteria consider genetic factors like the presence of the APOE ε4 genotype and specific blood biomarkers associated with AD. Studies have revealed that individuals with SCD often exhibit reduced gray matter volume in certain brain regions, particularly the hippocampus and frontal-temporal lobes. These changes in brain structure correlate with memory decline and cognitive impairment. The paper also discusses the use of resting-state functional magnetic resonance imaging (rs-fMRI) to examine brain activity patterns in individuals with SCD. Researchers aimed to understand how these brain activity patterns, measured by PerAF (persistent atrial fibrillation), relate to cognitive performance and blood biomarkers such as Aβ42 and Aβ40. In their study, the authors analyzed 64 individuals with preclinical AD and SCD, comparing them to 55 age-matched controls. Various assessments, including cognitive tests and blood tests, were conducted. The results showed differences in cognitive scores (MoCA) and blood biomarkers between the SCD group and the control group, suggesting that these factors might be linked to SCD. Additionally, the study found decreased gray matter volume in specific brain regions of patients with SCD, including the hippocampus, rectus gyrus, and precentral gyrus. The researchers also investigated changes in brain network properties, such as small-world properties and connectivity, and found alterations in these network characteristics in individuals with SCD. While there was a correlation between certain brain changes and cognitive decline, the study did not establish a clear link between these changes and blood biomarkers Aβ42 and Aβ40. In summary, the paper underscores the complexity of SCD and the importance of using various neuroimaging techniques to unravel its underlying mechanisms and its relationship with AD-related biomarkers. It suggests that SCD may involve both structural and functional brain changes, but further research with larger sample sizes is necessary to confirm these findings and provide a more comprehensive understanding of SCD in relation to Alzheimer's disease.
Overall, I find the objective presented in this article to be quite intriguing, and the authors' insightful observations on this relevant subject matter could capture the attention of Brain Sciences readers. However, there are certain points worth addressing, including specific comments and essential evidence required to bolster the author's argument. These adjustments are necessary to enhance the manuscript's quality, suitability, and overall readability before it can be published in its current state. In conclusion, I recommend publication of this research article once the author has thoughtfully incorporated the feedback and suggestions outlined below.
- The title is quite informative and appears to accurately convey the content of the research. However, here are a few suggestions for critique and potential improvements; for example, consider simplifying it for a broader audience while maintaining its accuracy. Also, the title could be made more concise by removing redundancy. For instance, "Changes in Blood Biomarkers" could be combined with "A Multimodal Imaging Analysis" to streamline the title.
- I strongly recommend to include a graphical abstract that summarizes the main findings of the manuscript.
- The Introduction section is generally clear and well-structured. However, it could be further improved by dividing it into more concise paragraphs, each focusing on a specific aspect of the research. I would also suggest to further elaborate on the specific brain imaging techniques authors plan to use in your study. Discuss their advantages and limitations. For example, in addition to resting-state fMRI, you could explore the potential of other imaging modalities such as structural MRI, PET scans, or diffusion tensor imaging (DTI) to provide a comprehensive view of brain changes associated with SCD, as well as further discuss structural changes in the brain, such as gray matter atrophy or white matter abnormalities, that have been associated with SCD. Finally, in my opinion, in would be beneficial to provide a brief overview of the neurobiological mechanisms that may underlie the observed changes in brain areas. For example, discuss the role of amyloid-beta and tau protein accumulation, neuroinflammation, or vascular factors in SCD and its progression to Alzheimer's disease [1-3].
- Regarding the clinical data collection, please provide more information about the specific neuropsychological scales used, including their reliability and validity. Also, here authors should mention who conducted these assessments and whether they were blinded to the participants' group status.
- In general, in the ‘Materials and Methods’ section, I recommend adding more details about Aβ42 and Aβ40 detection, consider adding more details about the Aβ42 and Aβ40 assay's sensitivity, specificity, and any quality control measures that were taken, as well as a better explanation of why APOE genotyping was performed and its relevance to the study.
- Preprocessing of Structural MRI Data: Can the authors provide more details about the criteria used to exclude participants based on image quality? Please explain why total intracranial volume (TIV) and education level were included as covariates.
- Finally, I believe that the discussion section appears well-structured and informative. It provides a detailed explanation of the study's findings and their implications. Still, while it is informative, some sentences are quite long and complex, which may make it challenging for readers to follow the arguments. Consider breaking down complex ideas into simpler sentences for better readability. Also, there is some repetition of points throughout the section. For example, it is mentioned the positive correlation between Aβ levels and cognitive decline multiple times, but these points can be consolidated to avoid redundancy.
- References: Authors should consider revising the bibliography, as there are several incorrect citations. Indeed, according to the Journal’s guidelines, they should provide the abbreviated journal name in italics, the year of publication in bold, the volume number in italics for all the references. Also, some of the references are out of date: please cite references from the last 10 years, particularly references from the recent 5 years.
I hope that, after careful revisions, the manuscript can meet the journal’s high standards for publication. I declare no conflict of interest regarding this manuscript.
Best regards,
Reviewer
References:
1. https://doi.org/10.3389/fnmol.2023.1217090
2. https://doi.org/10.3390/biomedicines11051248
3. DOI: 10.3390/biomedicines11030945
Comments on the Quality of English Language
Minor editing of English language required.
Author Response
We thank you for your thoughtful suggestions and insights. The manuscript has benefited from these insightful suggestions.
The manuscript has been rechecked and the necessary changes have been made in accordance with your suggestions. Changes are denoted in red text.
Responses to the Reviewers’ comments:
Question 1. Simplifying the title and reducing the length of sentences.
Reply: We thank the reviewer for the suggestion. We have made revisions to the title, background, and discussion sections to highlight the main focus of the article and reduce sentence complexity, thereby better emphasizing the theme of the paper.
Question 2. Including a graphical abstract.
Reply: We thank the reviewer for the suggestion. We have submitted a graphical abstract to provide a more visual representation of the main theme of the article.
Question 3. Including previous multimodal MRI studies.
Reply: We thank the reviewer for the suggestion. In this study, we have made improvements to the introduction section by providing an explanation of the disparities in DTI and structural magnetic resonance imaging findings among patients with SCD, as well as addressing the existing inconsistencies in current research. Moreover, we have employed PerAF as an alternative to ALFF and fALFF for analyzing the alterations in low-frequency signals in SCD patients.
Question 4. Providing more information about the specific neuropsychological scales used.
Reply: We thank the reviewer for the suggestion. We have provided additional details regarding the neuropsychological scales used in the study.The assessments were conducted by two neurologists who received professional training in neuropsychological assessment. They were blinded to the participants' group status during the assessment, ensuring objectivity.
Question 5. Elaborating on the measures taken to increase the sensitivity and specificity of Aβ42 and Aβ40 measurements, as well as explaining the relevance of APOE genotyping to the study.
Reply: We thank the reviewer for the suggestion. This method utilizes the specific reaction between antigens and antibodies, while enzyme technology is based on the color reaction between enzymes and substrates to display the binding of antigens and antibodies. This greatly enhances the sensitivity of detection, making the detection level comparable to radioactive assays. The colored products generated through enzyme reactions are mostly stable, which is beneficial for sample preservation. The absorbance (OD value) is measured at a wavelength of 450 nm using an ELISA reader, and the concentration of human Aβ42 or Aβ40 in the sample is calculated through a standard curve, resulting in more accurate results.
Being an APOE/ε4 carrier can potentially influence the relationship between the SCD characteristics and the presence of amyloid proteins. This study aimed to analyze these factors.
Question 6. Providing more details about the criteria used to exclude participants based on image quality and explaining why total intracranial volume (TIV) and education level were included as covariates.
Reply: We appreciate the reviewer's suggestion. To enhance the study's reproducibility, we have provided detailed supplementary information on the preprocessing of all participants' MRI scans. Additionally, we have added an evaluation of the image quality. The image quality of each subject is manually checked, and those who do not meet the standards are excluded, including obvious signal loss, artifacts, head motion artifacts, and other apparent image issues.
During the statistical analysis, total intracranial volume (TIV), education level, gender, and age were included as covariates to remove potential confounding factors. However, it is not appropriate to include this information in this section. We have moved this information to the section dedicated to data analysis and statistical methods.

Round 2
Reviewer 1 Report
Comments and Suggestions for Authors
All suggestions given to the authors were adopted and the paper is now much clearer and the methodology is more detailed for future studies that wanted to be based on this study. The reader will have a greater understanding of the subject. I consider it appropriate for publication.
Author Response
Thank you once again for your thoughtful suggestions and insights. The manuscript has greatly benefited from these valuable inputs.
Reviewer 2 Report
Comments and Suggestions for Authors
Dear Authors, I read the updated version of the manuscript. In my opinion, the paper has been improved; however, some points should be further changed.
1. The following sentence sounds misleading. “Several factors can influence serum Aβ levels. Buckley et al. found an association between age and an 315 increase in Aβ in patients with SCD carrying the ApoE4 gene [22, 38, 39], while lower educational level also significantly impacts Aβ elevation [3, 23, 40].“ Authors begins dealing with serum amyloid levels: however, most of the reported studies adopted amyloid PET to detect brain amyloidosis. Please reformulate the sentence. In general, along the text, I suggest to specify what type of measurement is used (i.e., serum, plasma, amyloid PET or CSF) whenever amyloid changes are discussed. Also, please verify the sentence “plasma phosphorylated tau (P-tau) and Aβ42/40 appear to be the best candidate biomarkers for symptoms of AD, including prodromal AD and dementia”. I recommend to also report the NIA-AA research framework guidelines for the in vivo AD biomarkers (i.e., CSF, amyloid and Tau-PET). In fact, blood biomarkers can show high performance and seems suitable in predicting amyloid positivity. However, it’s important to specify that current criteria contemplate specific markers (see Jack et al., 2018).
2. To this purpose, authors state: “However, our study did not find a significant correlation between small-world properties and Aβ42 or Aβ40. This inconsistency could be attributed to the different types of amyloid beta selected and the differences in participant selection.” With respect to the mentioned study (Ten Kate et al.), not only the type of amyloid beta (i.e., the isoform) but also the method for measurement beta amyloid changes (i.e., plasma versus amyloid PET) are different.
3. The following sentence should be changed. “A higher MMSE, MoCA, or ADL score represents better cognitive function.” While MMSE and MoCA are cognitive tests, ADL are scales for functional assessment. The instruments are different kinds of tools and cannot be considered within the same category. I suggest to separate the sentence, specifying that ADL high scores indicate better functional autonomy.
4. I suggest to briefly introduce the concept of “SCD-plus” as an attempt to define specific SCD features more at risk for decline.
Comments on the Quality of English LanguageNot relevant changes are needed.
Author Response
We thank you for your thoughtful suggestions and insights. The manuscript has benefited from these insightful suggestions.
The manuscript has been rechecked and the necessary changes have been made in accordance with your suggestions. Changes are denoted in red text.
Question 1. Authors begins dealing with serum amyloid levels: however, most of the reported studies adopted amyloid PET to detect brain amyloidosis.
Reply: We appreciate the reviewer's suggestion. The problem of inappropriate referencing in the literature has been specifically addressed through targeted revisions, and the arguments have been restated. Moreover, the origin of amyloid-beta protein has been explicitly identified, leading to a more coherent and lucid presentation of the arguments.
Question 2. With respect to the mentioned study (Ten Kate et al.), not only the type of amyloid beta (i.e., the isoform) but also the method for measurement beta amyloid changes (i.e., plasma versus amyloid PET) are different.
Reply: We appreciate the reviewer's suggestion. In order to enhance the rigor of the paper's expression, inappropriate statements in this section have been removed.
Question 3. The ADL scale, MMSE, and MoCA scale were introduced separately.
Reply: We thank the reviewer for the suggestion. I have made modifications to this section by providing separate introductions for the three scales: ADL scale, MMSE, and MoCA scale.
Question 4. briefly introduce the concept of “SCD-plus
Reply: We thank the reviewer for the suggestion. We provided a brief introduction to SCD-plus in the introduction section, which makes the article more comprehensive.
We once again express our gratitude for your thoughtful suggestions and insights.